# Differential Binding of ΔFN3 Proteins of *Bifidobacterium longum* GT15 and *Bifidobacterium bifidum* 791 to Cytokines Determined by Surface Plasmon Resonance and De Novo Molecular Modeling

**DOI:** 10.3390/ijms262110560

**Published:** 2025-10-30

**Authors:** Maria G. Alekseeva, Sophia S. Borisevich, Alfia R. Yusupova, Diana A. Reznikova, Dilara A. Mavletova, Andrey A. Nesterov, Margarita G. Ilyina, Natalia I. Akimova, Alexander A. Shtil, Valery N. Danilenko

**Affiliations:** 1Vavilov Institute of General Genetics, Russian Academy of Sciences, 119333 Moscow, Russia; alekseevamg@mail.ru (M.G.A.); alfia_yusupova@mail.ru (A.R.Y.); reznikova.da@phystech.su (D.A.R.); mavletova@vigg.ru (D.A.M.); andrey0012345@mail.ru (A.A.N.); valerid@vigg.ru (V.N.D.); 2Synchrotron Radiation Facility—Siberian Circular Photon Source “SKlF” Boreskov Institute of Catalysis of Siberian Branch of the Russian Academy of Sciences, 630559 Koltsovo, Russia; sophiamonrel@gmail.com (S.S.B.); m.g.ilina@srf-skif.ru (M.G.I.); 3Moscow Center for Advanced Studies, 123592 Moscow, Russia; 4Blokhin National Medical Research Center of Oncology, 115522 Moscow, Russia

**Keywords:** intestinal microbiota, bifidobacteria, fibronectin binding domains, recombinant ΔFN3 proteins, protein-protein interactions, cytokines, tumor necrosis factor α, gene expression

## Abstract

Bifidobacteria, a genus of obligate anaerobes, comprise a major component of the intestinal microbiota. Importantly, bifidobacteria participate in immune reactions. These bacteria carry a species-specific operon in which the *fn3* gene encodes a multifunctional protein FN3 that mediates bacterial adhesion to the intestinal epithelium and is capable of binding individual cytokines. Bioinformatics and biochemical approaches were used to study the possible interaction of recombinant ∆FN3 fragments of *B. longum* and *B. bifidum* strains with cytokines TNF-α, IL-6, IL-8, and IL-10. De novo molecular modeling generated, for the first time, the structural models of species-derived ∆FN3 proteins and revealed new tentative regions for differential cytokine binding. Combined treatment with ∆FN3 and TNF-α induced TNF-α mRNA abundance in the human monocytic cell line. Altogether, these findings provide structural evidence for the regulation of immune reactions by microbiota-derived proteins.

## 1. Introduction

Intestinal microbiota emerges as a key factor of homeostasis [1]. Commensal bacteria, in particular, the *Bifidobacterium* genus, gained momentum as functional modulators of a variety of physiological processes as well as in disease [1,2]. The *B. longum infantis* and *B. bifidum* species are the first bacteria that colonize the intestine of newborns [3,4], suggesting a principal role of microbiota in post-embryonic development. Of particular interest is a cross-talk between the intestinal microbiota and the immune system [5,6,7]. Liwen et al. (2018) demonstrated a role for bifidobacteria in the balance of cytokines TNF-α/IL-4 and IL-5/IL-17A as a tentative biomarker of asthma in children [8]. Furthermore, the addition of bifidobacteria to the cultured colon carcinoma cell line decreased the expression of TNF-α, IL-6, and IL-12 and concomitant elevation of IL-10 and IL-8 mRNAs [9]. Thus, the outcome of bifidobacteria–cytokine interactions may vary from pro- to anti-inflammatory depending on the specific context [10].

Mechanisms whereby bifidobacteria and cytokines interact deserve an in-depth investigation. In particular, these interactions presume direct binding of cytokines to bifidobacterial proteins. Can bifidobacteria provide candidate protein binders that physically target (selectively or promiscuously) mammalian cytokines and modulate their functions? Previously, we discovered a species-specific PFNA operon that determines a significant diversity of amino acid sequences between strains of different species of bifidobacteria. The operon contains five genes: *pkb2*, *aaatp*, *duf58*, *tgm*, and *fn3* [11,12] that encode, respectively, the serine-threonine protein kinase [11,13,14]; MoxR-ATPase AAA-ATP, a protein with an unknown function that contains the annotated DUF58 domain, and a predicted transglutaminase [15]. The *fn3* gene encodes a ~210 kDa membrane-bound FN3 protein. The functional roles of this protein in bifidobacteria are poorly understood. In *B. bifidum* strain S17, FN3 mediated the adhesion of bacteria to the intestinal epithelium [16]. Thus, FN3 proteins can mediate protein–protein interactions, and specifically, bind the recombinant cytokines in cell-free systems [15,17].

Bifidobacterial FN3 proteins contain two fibronectin type III domains (FNIII, FN3). These domains were initially identified in fibronectin, a component of the extracellular matrix [18,19,20], where they act as structural spacers or are involved in protein–protein interactions [21]. FN3 domains are structurally close to immunoglobulin domains with seven β-sheets, but the intradomain disulfide bond in FN3 is absent [22,23]. Nevertheless, FN3 domains are conformationally stable, thereby providing the protein binding motifs different from those engaged by antibodies [24]. Generally, FN3 domains share a consensus motif WSXWS (WS motif) in which the conservative residues WS…WS have been reported in the type I cytokine (e.g., IL-21) receptors [25]. Mutations in the WS motif affected interaction with cytokines [26,27]. The residue X plays a role in the spatial organization of protein–protein complexes and in signaling [25,28,29]. These data mechanistically attributed the WS motif in the binding of cytokines to their cognate receptors.

Our comparative analysis of nucleotide sequences of bifidobacterial FN3 domains revealed species-specific variations of the annotated WS motif: WSXPS, WSXES, WSXDS, or WSXYS. The most pronounced differences were found in *B. angulatum* and *B. bifidum*: the WS motif in the 2nd domain of FN3 proteins was absent [15,30]. One may hypothesize that the above stretches can be involved in the interaction of bifidobacteria with cytokines.

The FN3 protein in the *B. longum* GT15 strain carries two tandem domains. The region 3′ to the 2nd domain is the C-terminus [31]. Previously, we cloned the *fn3* gene fragment encoding both FN3 domains and the C-terminal region (aa. 1494–1994). The protein expressed from this construct has been termed ΔFN3.1, where the symbol “Δ” denotes a truncated recombinant fragment encompassing two FN3 domains and the C-terminal region (aa. 1494–1994). This notation is used solely to distinguish the recombinant fragment from the full-length membrane-bound FN3 protein (~210 kDa) [17,31]. We demonstrated that ∆FN3.1 of *B. longum* GT15 can bind the recombinant TNF-α as determined in enzyme-linked immunosorbent assays [17]. This binding requires full-length ∆FN3.1, including the C-terminal region. The initial in silico analysis suggested that ΔFN3.1 can form a cavity for TNF-α binding [31]. Furthermore, treatment of *B. longum* GT15 with tumor necrosis factor α (TNF-α) modulated transcription of 176 operons (~1000 genes). Importantly, this treatment activated the expression of PFNA genes, including *fn3* [32]. These results provided strong evidence in favor of the link between bifidobacterial FN3 proteins and cytokines, thereby justifying an in-depth analysis of structural and functional interactions between bifidobacteria and the immune system.

Recent advances in de novo protein structure prediction have expanded the toolbox for structural biology. Among the most powerful methods are deep learning-based approaches such as AlphaFold2 [33] and RoseTTAFold [34]. The former approach demonstrates an exceptional accuracy in predicting global folds of single and multi-domain proteins, especially when evolutionary information is abundant. RoseTTAFold provides superior modeling of inter-domain arrangements and flexible linkers due to its three-track neural network architecture that simultaneously processes sequence, distance, and coordinate information. In contrast, I-TASSER [35,36,37] relies on threading and fragment assembly, which may limit its performance for proteins lacking close homologs. Given the absence of experimentally resolved structures for bifidobacterial FN3 proteins, we employed a multi-platform strategy to ensure the robustness of our models and to cross-validate the predictions.

In the present study, we expanded the array of bifidobacterial strains and cloned the genes encoding ΔFN3 in *B. angulatum* и *B. bifidum*. Comparison of the binding of recombinant ΔFN3 proteins from different species of bifidobacteria to the cytokines TNF-α, IL-6, IL-8, and IL-10 demonstrated differential affinity to individual cytokines, as determined by the surface plasmon resonance technique. We, for the first time, predicted in silico tertiary structures of ΔFN3 fragments and identified new potential cytokine binding regions. In cell culture, the combined treatment with FN3 and TNF-α elevated the abundance of TNF-α mRNA, suggesting that the consequences of interactions between microbiota and the immune system should be considered with caution.

## 2. Results

### 2.1. Analysis of Amino Acid Sequences of Bifidobacterial ΔFN3 Fragments

The ΔFN3 protein in *B. angulatum* GT102 (NCBI AMK57067, locus_tag Bang102_000210; 1956 aa. residues; GenBank GCA_000966445.2) [30] contains two domains FN3 (residues 1454–1539 and 1544–1635). The ΔFN3 protein in *B. bifidum* 791 (NCBI KYJ84759, locus_tag APS66_RS03355; 1992 residues; GenBank GCA_022014355.1) also contains two domains FN3 (residues 1490–1576 and 1581–1665). The protein expressed from this DNA fragment was termed ΔFN3.1. The names of respective proteins in *B. angulatum* GT102 and *B. bifidum* 791 strains were ΔFN3.2 and ΔFN3.3. Amino acid sequence identities (similarities) were: 33 (49%) between ΔFN3.1 and ΔFN3.2; 39 (54%) between ΔFN3.1 and ΔFN3.3; and 32 (46%) between ΔFN3.2 and ΔFN3.3.

Alignment of amino acid sequences of bifidobacterial ΔFN3 proteins revealed motifs other than the previously described conservative WSXWS stretch. In *B. longum,* the 1st FN3 domain contains the WSXPS sequence in which the tryptophan residue is substituted with proline. In the 2nd domain, the respective motif is WSXES (glutamic acid instead of tryptophan). The most pronounced difference was detectable between the 2nd FN3 domains of *B. angulatum*: SGXXAS (W → S, S → G, W → Q, S → A) and *B. bifidum*: EGXPS (W → E, S → G, W → P).

Alignment of amino acid sequences of ΔFN3.1 *B. longum* GT15, ΔFN3.2 *B. angulatum* GT102, and ΔFN3.3 *B. bifidum* 791 revealed that the homologies in domains 1 were more pronounced than in domains 2 (Figure 1). Thirty-five core (conservative) residues were detected in the 1st domain compared to 16 in the 2nd domain. The C-terminal region is the most variable among bifidobacteria. Therefore, expanding the array of bifidobacterial strains to investigate the binding of species-specific FN3 proteins with cytokines, we cloned the genes encoding ΔFN3.2 *B. angulatum* and ΔFN3.3 *B. bifidum,* isolated the respective proteins, and compared their characteristics with the previously obtained ΔFN3.1.

### 2.2. Cloning and Expression of Genes Encoding ΔFN3.2 B. angulatum and ΔFN3.3 B. bifidum Isolation and Purification of Recombinant Proteins

Previously, we cloned the *fn3* gene fragment (residues 1494–1994) of *B. longum* GT15 that encompasses two FN3 domains and a C-terminal region. The recombinant ΔFN3.1 was isolated [17,31]. In the present study, we cloned the genes encoding ΔFN3.2 *B. angulatum* and ΔFN3.3 *B. bifidum*. For PCR amplification of fragments of genomic DNAs of *B. angulatum* GT102 and *B. bifidum* 791, the oligonucleotides homologous to N- and C-terminal regions of the *Δfn3.2* and *Δfn3.3* genes, respectively, were used. The amplified fragments were cloned into the pET16b expression vector, followed by the procedures of protein expression and purification (see Section 4). SDS-PAGE showed ~56 kDa bands that corresponded to the calculated molecular masses of ΔFN3.2 and ΔFN3.3, including the His-Tag linker of the pET16b plasmid (Appendix A).

The criterion for selecting proteins for further research is the possibility of obtaining target proteins in preparative quantities in a native and homogeneous state. To test the possibility of purification of ΔFN3.2 *B. angulatum* and ΔFN3.3 *B. bifidum* under native conditions, pellets of *E. coli* BL21 (DE3) containing recombinant plasmids were lysed in a buffer (50 mM NaH_2_PO_4_, 5 mM Tris-HCl, 300 mM NaCl, 1 mM PMSF, 5 mM DTT (pH 8.0) containing lysozyme (1 mg/mL), 0.5% Triton X-100, and 20 mM 2-mercaptoethanol, sonicated with ultrasound and centrifuged to separate soluble fraction and the pellet. SDS-PAGE revealed that *B. bifidum* ΔFN3.2 is soluble, allowing for its purification under native conditions. The *B. angulatum* ΔFN3.3 protein was found in the pellet (inclusion bodies) and can be isolated only under denaturing conditions and subsequent re-folding. Thus, for further studies, ΔFN3.3 *B. bifidum* were used. The recombinant protein ΔFN3.3 of *B. bifidum* was purification in preparative quantities, the purity of the isolated protein was confirmed using SDS-PAGE (Appendix A). 

Two approaches were used to investigate the physical interaction of ΔFN3.1 *B. longum* and ΔFN3.3 *B. bifidum* with cytokines: a kinetic analysis by surface plasmon resonance and in silico simulations. To conduct a comparative kinetic analysis of interactions with cytokines, a fresh recombinant protein ∆FN3.1 of *B. longum* GT15 was also purificated.

### 2.3. Interactions of ΔFN3.1 B. longum GT15 and ΔFN3.3 B. bifidum 791 with TNF-α, IL-6, IL-8, and IL-10

We determined the ability of ΔFN3.1 *B. longum* GT15 and ΔFN3.3 *B. bifidum* 791 to bind the recombinant TNF-α, IL-6, IL-8, and IL-10 using the surface plasmon resonance technique. These analyses were performed in neutral (pH 7.4) or alkaline (pH 8.0) solutions. These values were chosen based on the isoelectric points of ΔFN3.1 *B. longum* (pI 6.05) и ΔFN3.3 *B. bifidum* (pI 7.18). As shown in Table 1 and Appendix A, ΔFN3.1 *B. longum* GT15 and ΔFN3.3 *B. bifidum* 791 were avid TNF-α binders: dissociation constants were 13.1 nM and 58.2 nM, respectively. However, the efficacy of binding to interleukins was different. For ΔFN3.1 *B. longum* GT15, pH did not influence the binding to IL-8 (Table 2, Appendix A). In contrast, at pH 7.4, the binding of ΔFN3.1 *B. longum* GT15 to IL-10 (Table 2, Appendix A) was detectable (K_b_ = 62.2 nM). The ΔFN3.1 *B. longum* GT15 interacted with IL-6 neither at pH 8.0 nor at pH 7.4 (Table 2, Appendix A). Furthermore, binding of ΔFN3.3 *B. bifidum* 791 to IL-8 (Appendix A), IL-10 (Appendix A), and IL-6 (Appendix A) did not depend on pH (Table 3). Interactions of ΔFN3.3 *B. bifidum* 791 with IL-6 and IL-10 at either pH were below the level of detection. Thus, binding of ΔFN3.1 *B. longum* GT15 with IL-10 was pH-sensitive, probably due to conformational changes at alkaline conditions. ΔFN3.3 *B. bifidum* 791 showed no detectable interaction with IL-10.

These results indicated that ΔFN3.1 *B. longum* GT15 and ΔFN3.3 *B. bifidum* 791 differentially bind to TNF-α, IL-6, IL-8, and IL-10. We next performed a systematic analysis of protein–protein interactions using molecular modeling, aiming to identify critical parameters of these interactions.

### 2.4. Prediction of Tertiary Structures

Prior to our study, experimentally confirmed 3D structures of bifidobacterial ∆FN3.1 and ∆FN3.3 proteins were not received. Therefore, we performed de novo structural prediction using state-of-the-art deep learning servers: AlphaFold2, RoseTTAFold, I-TASSER, and IntFOLD5 [38,39]. Importantly, no homology modeling was performed, as suitable structural templates with high sequence identity were absent [31]. (Figure 2A). In so doing, we used data on eukaryotic FN3 domain-containing proteins resolved by X-ray crystallography. Presumably, the proteins are formed by two linear FN3 domains with the predominant anti-parallel β-sheets.

Tertiary structures of ΔFN3 proteins obtained with RoseTTAFold were used as references for comparison with the results of AlphaFold2 and IntFOLD5 (Appendix A). For MD simulations and refinement of geometrical parameters, ∆FN3.1-AF2 and ∆FN3.1-IF5 models were chosen. The absence of black triangles in the prohibited areas reflects the absence of steric conflicts, that is, the models demonstrated high probability, similarly to RoseTTAFold.

Geometrical parameters of tertiary structures were refined for ∆FN3.1, predicted with RoseTTAFold (∆FN3.1-RF) and AlphaFold2 (∆FN3.1-AF2), as well as for ∆FN3.3, predicted with RoseTTAFold (∆FN3.3-RF) and IntFOLD5 (∆FN3.3-IF5). MD simulations lasted for 100 ns, followed by evaluation of RMSD fluctuations, clusterization, and Ramachandran map analyses of the most representative structures. MD simulations of ∆FN3.1-RF and ∆FN3.1-AF2 were completed successfully and reached the plateau (see RMSD curves in Appendix A). The amplitude of RMSD fluctuations for ∆FN3.1-AF2 was substantially lower than for ∆FN3.1-RF, suggesting that the latter model is less stable. Evaluation of the spatial organization of representative frames using Ramachandran maps (Appendix A) revealed a large number of sterically unfavorable conformations of the side chains in the amino acid residues in ∆FN3.1-RF compared to ∆FN3.1-AF2. Therefore, the latter model was considered in subsequent calculations.

The tertiary model of ∆FN3.3-RF achieved an equilibrium state neither within the initial 100 ns of MD simulations nor after an additional 100 ns (Appendix A). To ensure that MD trajectories reached equilibrium, we applied the quantitative criteria: (i) the backbone RMSD plateaued (fluctuations ≤ 0.2 Å over the last 20 ns), (ii) the radius of gyration stabilized, and (iii) the Ramachandran outliers remained below 1%. For ∆FN3.3-RF, RMSD continued to drift beyond 100 ns (Appendix A), indicating a failure to equilibrate; this model was excluded from further analysis. In contrast, ∆FN3.1-AF2 and ∆FN3.3-IF5 achieved stable RMSD plateaus (1.8 ± 0.3 Å and 10.5 ± 1.2 Å, respectively), confirming convergence. Clusterization of trajectory data was not performed. MD simulations of the ∆FN3.3-IF5 structure, followed by the clusterization procedure, generated a tertiary structure for molecular modeling stable for 100 ns. The RMSD curve (Appendix A) calculated by Cα atoms reached a plateau and fluctuated within 8–12 Å. The Ramachandran maps (Appendix A) revealed minor steric conflicts and irregular conformations in the loops and terminal zones. Nevertheless, in the clusterization procedure, the representative frames reflected the most stable conformations. The tertiary structure of ΔFN3.3 was chosen based on these data.

Thus, we focused on tertiary structures of ΔFN3.1 and ΔFN3.3 proteins obtained with AlphaFold2 and IntFOLD5, respectively. Of note, the models depicted in Figure 2B,C differed from the reported tertiary structures [31]. The reasons for these discrepancies remain to be elucidated; however, all three methodologies of structural prediction generated close results (Appendix A).

Geometrical parameters of TNF-α, IL-8, IL-10, and IL-6 were taken from PDB. Among 37 structures [40], 4G3Y [41] and 5UUI [42] fit the parameters obtained by X-ray analyses with the best resolution. For interleukins, the best geometrical parameters corresponded to the PBD codes 1ALU [43]—IL-6, 5D14—IL-8, and 2H24 [44]—IL-10. Also, we used an AlphaFold2 methodology for the prediction of tertiary structures of the above proteins. Since these proteins are well studied, the results of the predictions were good (Appendix A). Unlike for ΔFN3 proteins, different servers were unnecessary. Geometrical parameters of predicted tertiary structures of TNF-α, IL-8, IL-10, and IL-6 were refined during 100 ns MD simulations. The analysis produced the geometrical parameters of proteins that corresponded to the most representative clusterization frames (Appendix A).

### 2.5. Molecular Docking

#### 2.5.1. FN3-TNF-α Interaction

The region(s) in TNF-α involved in the interactions with ∆FN3 proteins has not been identified [31]. For FN3 domain-containing eukaryotic proteins, the interactions take place in the cytokine receptor motif. We hypothesized that in the ΔFN3.1 protein of *B. longum* GT15, the following amino acid residues may be involved in binding to cytokines: Trp78, Ser79, Pro81, and Ser82 (the “cytokine receptor motif” in the 1st domain of FN3, annotated in NCBI); Trp174, Ser175, Glu177, Ser178 (the “cytokine receptor motif” in the 2nd domain of FN3, annotated in NCBI), and Ala43, Ala51, Thr111, Pro417, and Ala424, which we have identified in 203 sequenced genomes of *B. longum* subsp. *longum* [31]. For the *B. bifidum* 791 ΔFN3.3, it can be assumed that the residues involved in cytokine binding are located in the cytokine receptor motifs in the 1st and 2nd domains (Table 4). The amino acid sequences of ΔFN3.3 proteins of all *B. bifidum* genomes were identical (our unpublished data).

Protein–protein docking procedures generated 30 poses of the cytokine relative to FN3 proteins. In the first approximation, we selected optimal docking poses (Appendix A). Selection was based on energy parameters and the analysis of intermolecular interactions between the amino acid residues in the cytokine and FN3. Also, we considered the involvement of residues presented in Table 4. Then, two poses were chosen for MD refinement of the geometrical parameters of ΔFN3.1-TNF-α and ΔFN3.3-TNF-α complexes.

MD simulations of ΔFN3.1-TNF-α complexes in positions 24 and 5, as well as ΔFN3.3-TNF-α complexes in positions 21 and 14, were performed for 200 ns. RMSD fluctuations are given in Appendix A. The ΔFN3.1-TNF-α-pose5 equilibrated by 120 ns (Appendix A), unlike ΔFN3.1-TNF-α-pose24 (Appendix A). RMSD fluctuations were ~10 Å. A similar conclusion can be drawn from the analysis of ΔFN3.1-TNF-α MD curves (Appendix A): in the ΔFN3.3-TNF-α-pose14, RMSD fluctuations of ΔFN3.3 and TNF-α were within 2–4 Å by the completion of simulations (Appendix A), whereas in ΔFN3.3-TNF-α-pose21, the fluctuations were significant (Appendix A).

Clusterization procedures produced a representative frame that corresponded to the most stable position of the system over the entire time of simulations. This frame contains similar RMSD values. In all cases, Ramachandran maps were good, with a few insignificant steric conflicts (Appendix A). Free energies of binding (ΔG_bind_) and dissociation constants (K_D_) were estimated for each complex. Since the binding regions of TNF-α and ΔFN3 are uncertain, we selected the complexes with K_D_ values close to experimental values (Appendix A).

During the entire time of simulation, TNF-α localized predominantly between epitopes I and II of ∆FN3.1 and formed intermolecular interactions such as hydrogen bonds, salt bridges, and π–π stacking (Figure 3A, Appendix A). In the representative frame ∆FN3.1-TNF-α-pose24), the hydrogen bonds were found between the residues in TNF-α and epitope II; residues W174-Q123, D158-S147, and N173-Q123 were critically important (Appendix A). In pose 5, the cytokine was bound to FN3. Intermolecular contacts between Y195 in TNF-α formed π–π cation stacking bonds with W174 (Figure 3B). This position of the cytokine relative to ∆FN3.1 was characterized by a smaller number of hydrogen bonds compared to pose24. However, the theoretically calculated K_D_ value was closer to the experimental data.

Most probably, TNF-α interacts with ∆FN3.3 between the epitopes. Nevertheless, numerous intermolecular contacts with epitope III were registered (Appendix A, Figure 3C). In pose14, the pairs of residues in the complexes form hydrogen bonds (Figure 3D). In pose21, the salt bridges were formed between K204-E219 in ∆FN3.3 and K216-E180 in the cytokine (Appendix A). Values ΔG_bind_ for TNF-α-∆FN3.3 complexes in pose14 were below those in pose21. Consequently, K_D_ (∆FN3.3-TNF-α-pose14) was comparable with experimental data (Appendix A, Figure 3D). We suggested that the main differences between K_D_ values of TNF-α-FN3 complexes are dictated by the differential positioning of TNF-α relative to the surfaces of ∆FN3.1 and ∆FN3.3.

#### 2.5.2. FN3–Interleukin Interactions

The potential binding site(s) of bifidobacterial ∆FN3 proteins with interleukins is also unknown. We selected optimal docking poses out of 30 for MD-assisted refinement of geometrical parameters of ∆FN3-interleukin complexes (Appendix A). For IL-8, two poses were selected in which the cytokine was located between the epitopes I and II (pose3) or II and III (pose11) in ΔFN3.1. A number of hydrogen bonds were found between the aa. residues (Appendix A). In the majority of docking poses, the contacts of ∆FN3.3 with IL-8 involved the epitope V. In this case, we analyzed two opposite poses: #17 (IL-8 bound to the epitope V) and #19 (IL-8 bound to the epitope III; Appendix A). In the latter pose, only a few hydrogen bridges were detected, unlike ΔFN3.3-IL-8 (pose17). The analysis of MD trajectories showed that both ΔFN3.1-IL-8 (pose3) and ΔFN3.1-IL-8 (pose11) reached an equilibrium by the end of the simulation (Appendix A). Still, for ΔFN3.1-IL-8 (pose11) complexes, RMSD fluctuations of IL-8 atoms were more pronounced (~10 Å; Appendix A) than in ΔFN3.1-IL-8 (pose3) complexes (Appendix A). The interaction of IL-8 with the C-terminus of ΔFN3.3 (pose17) was more stable than with epitope III (pose19; Appendix A). The Ramachandran maps that corresponded to the representative frames were satisfactory (Appendix A). The calculated values ΔG_bind_ and K_D_ suggested that the probable positioning of IL-8 on the surface of ΔFN3.1 and ΔFN3.3 corresponded to pose3 (Figure 4A) and pose17 (Figure 4B), respectively. The calculated K_D_ values were in agreement with experimental values.

In the case of IL-10, we selected two poses: IL-10 binds to the C-terminus of ΔFN3.1 (pose16) or the cytokine is localized between the epitopes I and II (pose28). In the former scenario, more intermolecular hydrogen bonds and salt bridges were registered compared with pose28 (Appendix A). One may suppose that IL-10 interacts with ΔFN3.3 via the C-terminus (Appendix A). Each of four MD simulations showed an unstable positioning of the interleukin relative to the surface of ΔFN3 proteins (Appendix A). Representative frames with Ramachandran maps were obtained only for ΔFN3.1-IL-10 complexes in two poses (Appendix A). Values ΔG_bind_ for both poses were comparable (Appendix A). Nevertheless, K_D_ values suggest that, predominantly, IL-10 makes contact with the epitope V at the C-terminus of ΔFN3.1 (Figure 3C).

According to the results of molecular docking, IL-6 preferentially binds to the C-termini of ΔFN3.1 and ΔFN3.3 (Appendix A), forming a small number of hydrogen bonds between the paired aa. residues. MD simulations revealed an unstable positioning of IL-6 relative to the surfaces of ΔFN3.1 or ΔFN3.3 (Appendix A). Since the experimental data showed a weak interaction of IL-6 with ΔFN3 proteins, we did not calculate ΔG_bind_ and K_D_ values.

### 2.6. Effects of ΔFN3 and TNF-α on Cytokine mRNA Abundance in THP-1 Cells

To obtain insight into the physiological significance of the binding of ΔFN3 to TNF-α, we studied the effects of the combination of these proteins on the expression of genes known to be regulated, at least in part, by TNF-α: *IL-6*, *IL-8*, and *TNF-α*. In so doing, we treated the THP-1 human monocytic leukemia cell line with ΔFN3 proteins alone or in combination with TNF-α for 3 h, followed by RT-PCR. Conditions of treatment were optimized in preliminary experiments. As shown in Figure 5, ΔFN3.1, ΔFN3.3, or TNF-α alone caused no significant increase in TNF-α mRNA except for relatively large (900 ng) amounts of ΔFN3 proteins. In contrast, the combinations of ΔFN3.1 or ΔFN3.3 with TNF-α elevated the abundance of TNF-α mRNA ~4–7-fold. The synergy was independent of the amounts of ΔFN3 proteins; that is, the fold increase of TNF-α mRNA was similar for each combination. These results strongly suggested that ΔFN3 can synergize with TNF-α in activating the gene encoding this cytokine. The effects of combinations were gene-specific: TNF-α alone was a strong inducer of the IL-8 mRNA, whereas no synergy with ΔFN3 proteins was detected (at least at the concentrations used in the experiments) (Appendix A). Also, no synergistic effect on the *IL-6* mRNA was observed.

## 3. Discussion

To date, the main components of the human immune system have been well studied and described [45,46,47]. These components include effector cells, receptors localized on them, and signaling molecules called cytokines, which regulate and control the organism’s immune homeostasis. Cytokines are the most important cellular signaling molecules that mediate the organism’s immune response [48]. Cytokines are divided into pro-inflammatory and anti-inflammatory. Pro-inflammatory cytokines play an important role in protecting organisms from pathogens. However, an excessive inflammatory response can lead to the development of chronic inflammation and disruption of homeostasis, which can cause diseases such as cancer, diabetes, cardiovascular diseases, and Parkinson’s disease [49]. The main proinflammatory cytokines include L-6, IL-8, and TNF-α. Based on its functions, the immune system is divided into innate and adaptive [50]. The adaptive immune system is believed to have emerged and functions in response to various effector factors, including pathogenic bacteria, viruses, and effector molecules of different natures.

The adaptive immune system plays an important role in self-or-foe recognition. Pathogenic bacteria can be classified as foreign agents. But to what category should commensal bacteria of the gut microbiota, and particularly bifidobacteria, be classified? Intensive research over the past decade into the human and animal gut microbiome has enabled the identification and characterization of commensal, functionally beneficial bacteria using effector molecules, proteins, and cellular components that interact with the host organism and, directly or indirectly, with components of its immune system [51,52,53].

Thus, the gut microbiome is another important component of the immune system. In addition to the microbiome–brain axis, the microbiome–immune system axis has begun to emerge and be studied. Then came the matter of searching for receptor sites within elements of this immune system component that are capable of interacting with the ligands and signaling molecules (cytokines) of the organism`s immune system. These receptor domains have been identified and partially characterized in bifidobacteria [12,15,17].

Bifidobacteria comprise an ancient group of anaerobic bacteria [54]. These microbes are the only representatives of actinobacteria in the gastrointestinal tract [55], accounting for 99 species [56]. *B. longum* and *B. bifidum* are the primary residents of the child’s intestine that participate in the immune system development [3]. Bifidobacteria and their metabolites have been implicated in intestine–brain and intestine–immunity cross-talks [5,57].

The FN3 protein, initially identified by us as a product of the PFNA operon, can be a candidate mediator of immune regulation by bifidobacteria. ΔFN3 proteins demonstrate a high interspecies divergence of amino acid sequences. Also, ΔFN3 carries various motifs of cytokine receptors; these structural features substantiate the differential affinity to certain cytokines. We have shown that the conservative cytokine receptor motifs in bifidobacteria contain WS-PS, WS-ES, WS-DS, or WS-YS stretches. The most striking was the difference between *B. angulatum* and *B. bifidum* [15], namely, the absence of the WSXWS motif in the 2nd FN3 domain.

In the present study, we dissected FN3-cytokine interactions in more detail. Previously, we have used ELISA to show that the FN3 fragment (∆FN3.1) of the *B. longum* GT15 strain preferentially binds TNF-α [17]. The structural model of ΔFN3.1 generated with the trRosetta software «https://yanglab.qd.sdu.edu.cn/trRosetta/ (accessed on 14 December 2021)» included five epitopes with β-sheets as predominant folds. Two epitopes are formed by two FN3 domains, whereas another three epitopes are presented in the C-terminal region [31]. We determined the binding profiles of ΔFN3.1 *B. longum* GT15 and ΔFN3.3 *B. bifidum* 791 with TNF-α, IL-6, IL-8, and IL-10 using the surface plasmon resonance measurements. Kinetic analysis of ΔFN3-cytokine interaction proved that ΔFN3.1 and ΔFN3.3 can bind TNF-α and IL-8. Furthermore, ΔFN3.1 was capable of binding IL-10, whereas IL-6 interacted with neither ΔFN3.

We next performed de novo molecular modeling [34] and neuronal network technologies to generate, for the first time, the tertiary structures of ΔFN3.1 and ΔFN3.3. Based on these models, we provided novel evidence regarding tentative cytokine binding regions in ΔFN3 proteins. The results of calculated binding energies and dissociation constants of ΔFN3-cytokine complexes were in good agreement with experimental data.

Most importantly, our modeling demonstrated that cytokine-binding regions differed for individual ΔFN3 proteins. For ΔFN3.1 *B. longum* GT15, TNF-α interacted presumably with the 2nd FN3 domain; this interaction involved aa. residues of the cytokine receptor motif WSXWS (Appendix A). In complexes ΔFN3.1-IL8 pose3 (Appendix A), the interaction involves W174 in the 2nd domain as well as the neighboring residues I172 and N173. In contrast, the above motif is absent in the 2nd domain of ΔFN3.3 *B. bifidum* 791; the binding involved other residues located in the 1st domain and the C-terminus. These findings, being not obviously predictable, expand our knowledge of mechanisms of interactions between microbiota and the immune system. Several protein regions in the participating proteins are involved, making these interactions non-incidental; mutational analysis will determine the significance of specific sites. One may hypothesize that individual regions provide more conformational opportunities that ultimately cooperate to form stable complexes. Thus, structural diversity of binding regions ensures the evolutionary conserved manner of microbiota–cytokine interactions.

Testing whether the affinity of ΔFN3 to cytokines in cell-free systems may be translated into the physiological effect, we observed that both ΔFN3.1 and ΔFN3.3 synergized with TNF-α in the activation of the TNF-α gene in the human monocyte cell line. We suggest that ΔFN3 binds TNF-α in the extracellular milieu, thereby facilitating the autoregulatory loop ΔFN3-TNF-α-TNF-α induction. The synergistic effect of the combination was specific for TNF-α since no additivity was registered for IL-8 or IL-6 mRNAs upon cell exposure to TNF-α together with either ΔFN3.1 or ΔFN3.3. These observations add complexity to the interpretation of the role of microbiota in immune regulation. TNF-α is a pleiotropic cytokine with established significance in a variety of immune reactions [58]. The gene encoding this factor carries two promoters with functionally opposite effects [59]. More studies are needed to clarify the mechanism(s) whereby the bifidobacterial proteins synergize with TNF-α to sustain TNF-α activation. One plausible explanation could be the acquisition of a specific confirmation of TNF-α in complexes with ΔFN3, making the cytokine a stronger inducer of the cognate gene. This assumption is in line with the above-mentioned conformational hypothesis: the more binding sites, the more variants of formation of functionally diverse protein–protein complexes. Finally, interference with the ΔFN3-TNF-α interaction with a small molecular weight compound or a peptide inhibitor might be considered for the disruption of the autoregulatory loop to attenuate stress-induced cytokine release.

Tumor necrosis factor TNF-α is a pleiotropic cytokine and one of the key modulators of the human immune system [58]. The data obtained do not allow us to definitively suggest the mechanisms of action that enhance TNF-α gene expression. According to the literature data, there is [59] a TNF-α expression pattern that takes into account the presence of two promoter regions, the preferential binding of which leads to opposite functional results. Further studies are required to establish the molecular mechanisms that determine the influence of the interaction of ΔFN3 proteins with TNF-α on the mechanism of regulation of its expression in certain effector immune cells of the human body.

IL-8 is an important inflammatory mediator that induces the expression of adhesion molecules on the vessel wall, in that way stimulating neutrophil binding, adhesion, and migration to tissue sites of injury, as well as inducing the formation of neutrophil extracellular traps (NETs) [60]. It is of great interest how the FN3.3 protein, through its interaction with IL-8, participates in these processes.

IL-6 is considered a major mediator of immune and inflammatory responses [48]. Furthermore, IL-6 is one of the key factors contributing to the virus-induced cytokine “storm.” IL-6 acts by binding to the membrane-bound IL-6R receptor and forming a complex with the membrane glycoprotein gp130. The latter step triggers the Jak/STAT and NFκB-mediated proinflammatory response in immune cells, such as macrophages, neutrophils, and T- and B-lymphocytes [61]. Tumor necrosis factor TNF-α is a pleiotropic cytokine and one of the key modulators of the human immune system.

The observed differences in cytokine-binding profiles between ∆FN3.1 (*B. longum*) and ∆FN3.3 (*B. bifidum*) may underlie species-specific immunomodulatory effects of bifidobacteria in the human gut. For instance, the ability of ∆FN3.1, but not of ∆FN3.3, to bind IL-10 (an anti-inflammatory cytokine), suggests that *B. longum* strains might contribute more effectively to the resolution of inflammation. Conversely, the stronger TNF-α binding by ∆FN3.1 could amplify pro-inflammatory signals, as demonstrated by upregulation of TNF-α mRNA in THP-1 cells. These findings highlight the potential of ∆FN3 proteins as strain-specific biomarkers for personalized probiotic selection and as scaffolds for engineered immunomodulators.

## 4. Materials and Methods

### 4.1. Bacterial Strains, Plasmid Vectors, Culture Media, and Conditions

We used the following strains: *B. angulatum* GT102 [30], *B. bifidum* 791, *E. coli* DH5a (F–, Φ 80 ΔlacZΔM15, Δ(lacZYA-argF), U169) (Promega, Madison, WI, USA) [62], and *E. coli* BL21(DE3) (F–, dcm, ompT, hsdS(r B–mB–), gal λ (DE3)) (Novagen, Madison, WI, USA). The expression vector pET16b (Novagen, Madison, WI, USA) [63] contains a His-Tag linker in the N-terminal region for protein isolation and purification. *B. angulatum* GT102 and *B. bifidum* 791 strains were cultured under anaerobic conditions (HiAnaerobic SystemeMark III, HiMedia, Maharashtra, Mumbai, India) in agar and MRS broth (HiMedia, Maharashtra, Mumbai, India) supplemented with cysteine (0.5 g/L) at 37 °C for 24–48 h. *E. coli* strains were propagated in Luria-Bertani (LB) broth [64]. Ampicillin (150 μg/mL) was used as a selective agent for the selection of plasmid-bearing cells. All reagents were from Amresco (Solon, OH, USA) unless specified otherwise.

### 4.2. DNA Manipulations

Genomic DNAs of *B. angulatum* GT102 and *B. bifidum* 791 were isolated using the GenElute Bacterial Genomic DNA Kit (Sigma-Aldrich, St. Louis, MO, USA). Isolation of plasmid DNA, obtaining competent *E. coli* cells, transformation, and analysis of recombinant plasmids were performed using standard methods [64]. DNA fragments carrying Δ*fn3* were amplified from genomic DNAs of *B. angulatum* GT102 and *B. bifidum* 791 strains using Phusion High Fidelity PCR Master Mix (Thermo Fisher Sci., Vilnius, Lithuania) on a mini-cycler PTC-0150 (MJ Research, Inc., Waltham, MA, USA). The following oligonucleotides were used for amplification: FN3B.ang-N: (5’-tcgtcatatgcccgacgccccgtcactgt-3’), FN3B.ang-C: (5’-gatcctcgagctaccgggaatacgtatgcaattc-3’), FN3B.bif-N(5’-tcgtcatatggacaagcccggcgcgccgc-3’), and FN3B.bif-C: (5’-gatcctcgagtcatggtcggtttgaggccag-3’) (all from Eurogen, Moscow, Russia). PCR-amplified fragments were cloned into the *Nde*I and *Xho*I restriction sites of the His-Tag-containing pET16b expression vector.

### 4.3. Expression in E. coli and Purification of Recombinant ΔFN3 Proteins

*E. coli* BL21 (DE3) cells containing the recombinant plasmid were grown in LB broth at 37 °C until they reached an OD_600_ of 0.6–0.8. The *fn3* gene of *B. angulatum* GT102 and *B. bifidum* 791 was induced by the addition of 1 mM isopropyl-β-D-1-thiogalactopyranoside (IPTG) for 5 h at 28 °C. Cells were pelleted and frozen at −20 °C. To study the expression of *fn3*, cells were resuspended in a sample buffer containing 62.5 mM Tris-HCl, pH 6.8, 5% glycerol, 2% 2-mercaptoethanol, 0.1% SDS, and 0.001% bromophenol blue, then heated at 95 °C for 10 min and analyzed by 12.5% SDS-PAGE. Protein fractions of *E. coli* BL21 (DE3) cells containing an empty pET16b plasmid were used as negative controls.

Isolation and purification of ΔFN3.1 *B. longum* GT15 and ΔFN3.3 *B. bifidum* 791 were carried out as described [17]. Samples were dialyzed in a buffer containing 10 mM HEPES-NaOH, pH7.4, 150 mM NaCl, 10% glycerol, and 1 mM phenylmethylsulfonyl fluoride (PMSF). Protein concentration was measured on a Qubit 2.0 fluorimeter (Invitrogen, Waltham, MA, USA). Purified proteins were stored at −80 °C.

### 4.4. Interaction of ΔFN3 Proteins with TNF-α, IL-8, IL-6, and IL-10

For surface plasmon resonance analysis (imSPR-Pro system, iCLIEBIO, Dangjin-si, Chungcheongnam-do, Republic of Korea), recombinant TNF-α, IL-8, IL-6, and IL-10 (ProSpec, Rehovot, Israel) were immobilized on the sensor COOH-chip (iCLIEBIO). Four different concentrations of the analyte sample (ΔFN3.1 *B. longum* GT15 or ΔFN3.3 *B. bifidum* 791) were prepared by serial dilution in a running buffer containing 10 mM HEPES-NaOH pH8.0 (or pH 7.4), 150 mM NaCl, and 0.05% Tween-20. The ΔFN3.1 *B. longum* GT15 or ΔFN3.3 *B. bifidum* 791 samples were injected at a flow rate of 30 µL/min. The injection step included a 200 s association phase followed by a 600 s dissociation phase. Serum albumin (SibEnzyme, Novosibirsk, Russia) was used as a control for non-specific protein binding. Data were analyzed by globally fitting curves describing the simple 1:1 bimolecular model to the set of three sensorograms using BIAEvaluation software version 4.1 «http://biaevaluation.software.informer.com (accessed on 4 February 2024)». Sensorograms characterize the changes of the signal over time. Signals were measured in resonance units (RU) proportionally to the amount of the surface-bound substance. Initially, the analytes ∆FN3 *B. longum* GT15 and *B. bifidum* 791 were loaded onto the chip and allowed to interact with TNF-α, IL-6, IL-8, and IL-10 for 200 s. The sensorograms registered the responses over time. The rate of complex association Ka (1/Ms) was calculated. Next, the chips were washed with a running buffer, and the complexes were allowed to dissociate for up to 600 s. This process reflected a decreased response on the sensorograms. The dissociation rate Kd (1/s) and dissociation constant K_D_ = K_d_/K_a_ (M) were calculated.

### 4.5. Molecular Modeling Studies

#### 4.5.1. Prediction of Tertiary Structure of ∆FN3 Proteins

Tertiary structures of ∆FN3.1 and ∆FN3.3 were predicted on the basis of AlphaFold2 [33], RoseTTAFold [34], I-TASSER [35,36,37], IntFold5 [38], and scoring functions [39]. In AlphaFold2 and RoseTTAFold, the accuracy of predictions is estimated as predicted Local Distance Difference Test (pLDDT) values that reflect local atomic distances [65]. The I-TASSER and IntFOLD5 tools use their evaluation functions to assess the accuracy of prediction, analogously to pLDDT: C-score (Confidence Score) [35,66,67] and GMQS (Global Model Quality Score) [38,68]. The algorithm is presented in Appendix A. Tertiary structures of TNF-α, IL-6, IL-8, and IL-10 were predicted using AlphaFold2 [33]. The geometrical parameters of each structure were refined by molecular dynamic (MD) simulations within 100 ns.

Selection of optimal structures among the array of predicted variants was based on scoring functions in AlphaFold2 [33], RoseTTAFold [34], I-TASSER [35,36,37], and IntFOLD5 [38]. Appendix A shows the quality of prediction for the best structures out of five variants. In all cases, the non-structured portions at the C-termini of ∆FN3.1 and ∆FN3.3 (aa. 475–503) were poorly predictable (red color). In general, the predictive results obtained with AlphaFold2, RoseTTAFold, and IntFOLD5 were similar. In contrast, the I-TASSER algorithm generated the structures with minimal relevance to the expected architecture of ΔFN3-containing proteins.

The analysis of Ramachandran maps (Appendix A) allowed us to estimate the geometrical probability of predicted ∆FN3 proteins. Top quality models were obtained with RoseTTAFold, AlphaFold2, and IntFOLD5: >90% amino acid residues were located in permissible areas (red and yellow) with a minimal number of prohibited conformations. I-TASSER generated the biggest number of geometrical alterations, making this algorithm not applicable for further analysis. Thus, we focused on predictions of ∆FN3.1 and ∆FN3.3 structures with RoseTTAFold. Recently, this server has been validated as preferred for the prediction of the full-size tertiary structure of poly(ADPribose) polymerase 1 [39].

#### 4.5.2. Protein–Protein Docking

To study the affinity of TNF-α, IL-8, IL-10, and IL-6 to FN3 proteins, we used the protocol of protein–protein docking PIPER [69]. The protocol is based on the method of quick Fourier transformation with a novel potential DARS (Decoys As the Reference State). The docking algorithm presumes the rotation of the ligand (TNF-α, IL-8, IL-10, or IL-6) relative to the receptor (∆FN3 protein) fixed in the coordinate system; all possible orientations of proteins relative to each other are detectable. The systems were considered solid bodies without optimization of the protein–protein interface. Because the binding regions have not been identified, the positions were unlimited. The docking poses were ranged according to the weight coefficients of energy terms PIPER score and PIPER energy [70].

#### 4.5.3. MD Simulations

Model systems of tertiary structures and protein–protein complexes were generated in the graphic milieu of the academic version of Maestro (Schrödinger Release 2024-1). The systems were placed into a cubic well with the buffer zone 15–25 Å filled with 0.15 M NaCl. Extra charges on proteins were neutralized with Na^+^ and Cl^−^. TIP3P was used as a solvent. MD modeling was performed in an NPT ensemble at 310 K (37 °C). Total time of simulations was 50–200 ns, 2 fs increment of the integrator, 5000–10,000 trajectory frames. The force field was OPLS4 [71]. MD simulations were performed using Desmond [72]. The analysis included mean square deviation of atomic positions (RMSD) and clusterization of frames [73].

#### 4.5.4. Estimation of Binding Energy and K_D_

Representative frames were aligned with geometrical parameters of statistically significant protein–protein complexes for which the free binding energies (∆G_bind_) were estimated according to MM-GBSA methodology [74]. Frames were selected based on minimal RMSD values and maximal number of repetitive images. The binding of TNF-α and interleukins to the surface of FN3 proteins can be expressed as:(1)$${\left[P+L\right]}_{eq}⇄{\left[P\right]}_{eq}+{\left[L\right]}_{eq},$$
where [P]_eq_ are FN3 proteins, [L]_eq_ are TNF-α or interleukins, [P+L]_eq_ is the protein–protein complex in an equilibrated state. The value ***∆G_bind_*** depends on the dissociation constants as:(2)$$\Delta G_{bind}=-RT\ln K_{D}^{{}^{\circ}},$$
where R is universal gas constant = 8.314 J/(mol × K), T is temperature, $K_{D}^{{}^{\circ}}$ is the dissociation constant:(3)$$K_{D}^{{}^{\circ}}=K_{i}^{-1}=\frac{\left[P+L\right]}{\left[P\right]\left[L\right]}.$$

The physical meaning of $K_{D}^{{}^{\circ}}$ represents the standard constant of equilibrium, a value transformed into K_D_ using the formula:(4)$$K_{D}=K_{D}^{{}^{\circ}}{\left(\frac{RT}{P}\right)}^{-∆\nu }$$
where P is pressure (H/m^2^), ∆ν is the sum of stoichiometry parameters of reaction coefficients (1).

### 4.6. Detection of Cytokine mRNA by Reverse Transcription-Polymerase Chain Reaction

The THP-1 human monocytic leukemia cell line (American Type Culture Collection, Manassas, VA, USA) was propagated in RPMI-1640 supplemented with 2 mM *L*-glutamine (PanEco, Moscow, Russia), 10% fetal bovine serum (Atlanta Biol., Flowery Branch, GA, USA), 50 U/mL penicillin, and 50 µg/mL streptomycin (PanEco, Moscow, Russia) at 37 °C, 5% CO_2_ in a humidified atmosphere. Cells in the logarithmic phase of growth were used in experiments. Cells were plated into 6-well plates (FDCELL, China; 4 × 10^5^/2 mL of culture medium) and treated with 300 ng of recombinant TNF-α (SCI STORE, Moscow, Russia) in the absence or presence of varying amounts of ΔFN3.1 or ΔFN3.3 for 3 h (Appendix A). Cells were pelleted, washed with saline, and lysed in ExtractRNA reagent (Evrogen, Moscow, Russia). Total RNA was isolated according to the manufacturer’s instructions. Reverse transcription was performed using the MMLV RT kit (Evrogen, Moscow, Russia). PCR mixtures (25 µL) contained 13 ng cDNA template, primers (0.4 µM each), qPCRmix-HS SYBR (Evrogen, Moscow, Russia), and deionized water. Amplifications were carried out on a CFX96 (Bio-Rad, Hercules, CA, USA) at 95 °C—5 min, 95 °C 30 s, 62 °C 30 s, and 72 °C 30 s (40 cycles). To analyze RT-PCR data, the CFX Manager V 3.1 program (Bio-Rad) was used. The *HPRT1* mRNA was taken as a reference [75]. Three biological replicates (each sample in triplicate) were analyzed; data were expressed as ∆∆Cq. Primers were designed using primer-BLAST [76] (Appendix A).

### 4.7. Bioinformatic Analysis

Nucleotide and amino acid sequences were from NCBI «http://www.ncbi.nlm.nih.gov/ (accessed on 13 December 2021)» and UniProt «http://www.uniprot.org/ (accessed on 17 December 2021)» databases. BLAST «https://blast.ncbi.nlm.nih.gov/Blast.cgi (accessed on 13 December 2021) » and Clustal Omega «https://www.ebi.ac.uk/jdispatcher/msa/clustalo (accessed on 14 December 2021)» programs were used for sequence alignment. Molecular weights and the isoelectric points of newly isolated proteins were calculated based on Molbiol servers «http://molbiol.ru/scripts/01_18.html (accessed on 17 January 2022)».

## 5. Conclusions

Interactions of the intestinal microbiota with the immune system form an important axis in normal homeostasis as well as in disease. The differential ability of individual bifidobacterial proteins to bind cytokines sets the stage for personalized use of chemical and biotechnological instruments that regulate these interactions. Knowledge about the molecular determinants of microbiota–cytokine binding can be expanded to other commensal species. Bifidobacterial ΔFN3 proteins are perspective as prototypic modulators of microbiome-immunity cross-talk, in particular, for the design of chemical or peptide disruptors for prophylaxis and therapy.

## Figures and Tables

**Figure 1 ijms-26-10560-f001:**
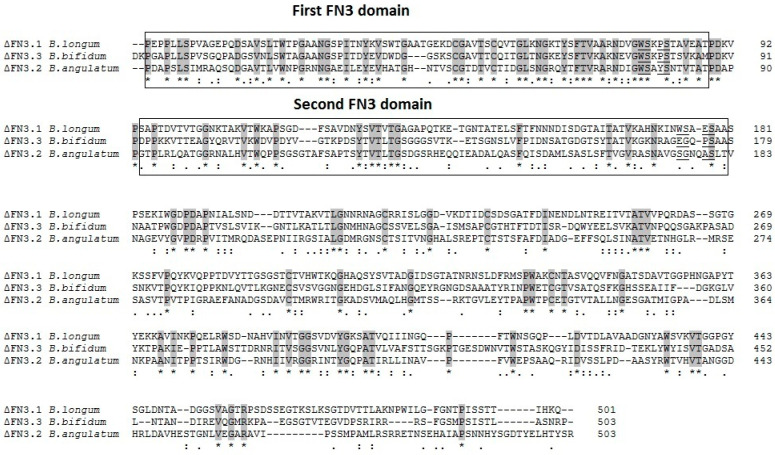
Alignment of amino acid sequences of ΔFN3.1 *B. longum* GT15, ΔFN3.2 *B. angulatum* GT102, and ΔFN3.3 *B. bifidum* 791. The conservative (core) residues are marked in gray, the annotated motifs of cytokine receptors are underlined. The FN3 protein domains are highlighted in the box. (*) - indicates matches of all three amino acids; (:) - two amino acids match, the third is similar; (.) - two amino acids match, the third is not similar.

**Figure 2 ijms-26-10560-f002:**
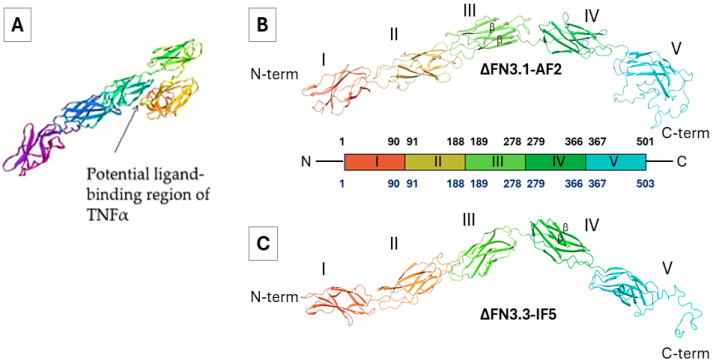
Tertiary architecture of ∆FN3.1 и ∆FN3.3 domains. (**A**): a V-shaped module of two linear FN3 domains carrying a potential ligand binding site (arrow; proposed in [31]; (**B**,**C**): tertiary structures of ΔFN3.1-AF2 и FN3.3-IF5 predicted by AF2 (AplhaFold2) and IF5 (IntFOLD5), respectively, and visualized as the chains of antiparallel β-sheets with numbered epitopes I–V. The colored architecture of proteins shows the numbered amino acid residues.

**Figure 3 ijms-26-10560-f003:**
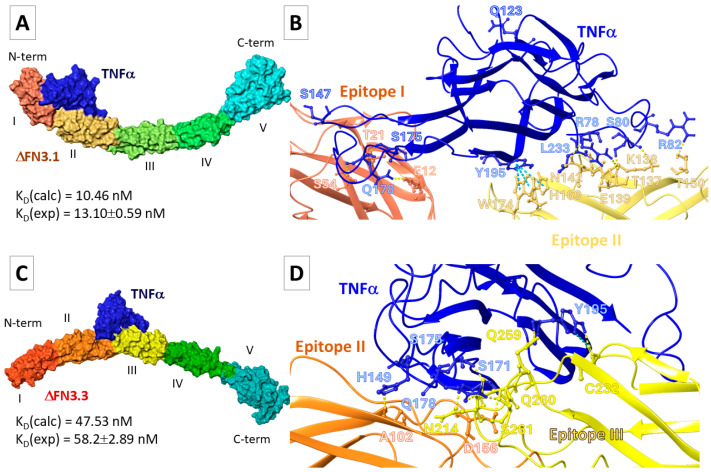
Results of MD simulations. Shown are the positions of TNF-α relative to ∆FN3.1 (**A**,**B**) and ∆FN3.3 (**C**,**D**) epitopes. Hydrogen bonds are rendered as dashed lines, π–π stacking interactions are depicted as blue lines. Colours indicated epitopes of ∆FN3 (I–V). The designations correspond to Figure 2.

**Figure 4 ijms-26-10560-f004:**
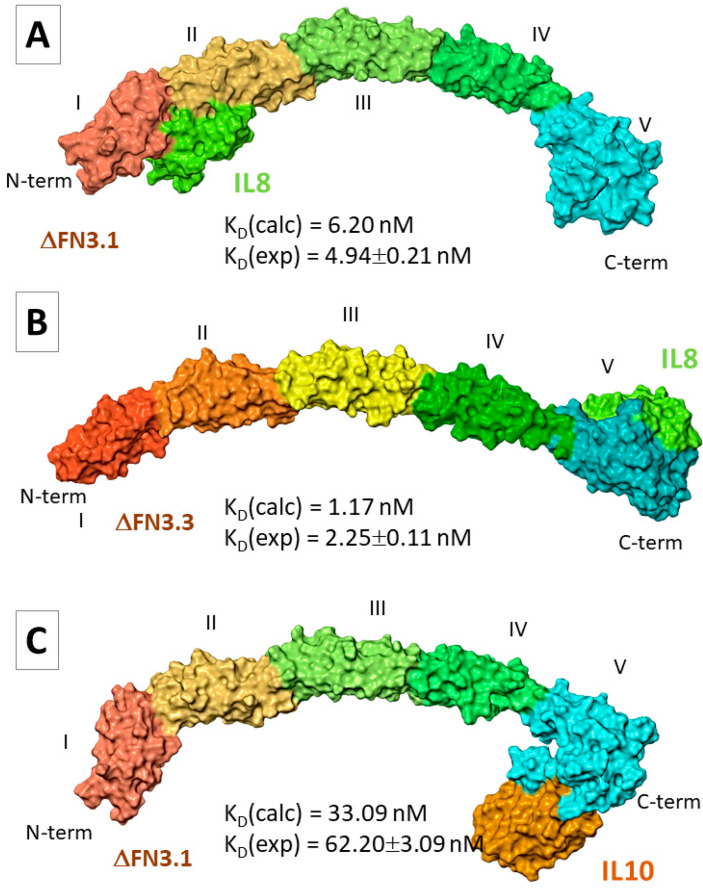
Molecular models of ΔFN3-interleukin complexes. (**A**) molecular model illustrating interaction ∆FN3.1 with IL-8; (**B**) molecular model illustrating interaction ∆FN3.3 with IL-8; (**C**) molecular model illustrating interaction ∆FN3.1 with IL-10. Colour indicate epitopes of ∆FN3 (I–V). IL-8 is indicated in light green. IL-10 is indicated in mustard colour.

**Figure 5 ijms-26-10560-f005:**
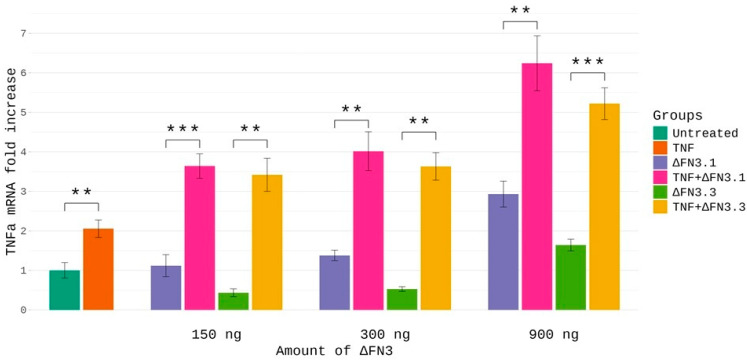
Synergistic combinations of ΔFN3 proteins and TNF-α in regard to TNF-α mRNA abundance. THP-1 cells were treated with indicated concentrations of ΔFN3.1, ΔFN3.3, or 300 ng TNF-α (alone or in combinations) for 3 h, followed by total RNA isolation and RT-PCR. Gene-specific signals were normalized on *HPRT1* cDNA. Each sample was analyzed in triplicate. Shown are mean + S.E.M. of three biological replicates. The mRNA levels in untreated cells were taken as 1. Asterisks *p* < 0.01 (**), or *p* < 0.001 (***) between the groups ‘combination’ vs. each protein alone. An independent Student’s *t*-test was used for statistical analysis.

**Table 1 ijms-26-10560-t001:** Parameters of binding of ΔFN3.1 *B. longum* GT15 and ΔFN3.3 *B. bifidum* 791 to TNF-α.

TNFα	K_a_ (1/Ms) *	K_d_ (1/s)	KD (nM)
ΔFN3.1	3.43 × 10^−5^	451 × 10^−5^	13.1 ± 0.6
ΔFN3.3	13.1 × 10^−5^	76.1 × 10^−5^	58.2 ± 2.9
Albumin	Below the level of detection		

* Binding parameters were calculated based on sensorograms using BIAEvaluation program Version 4.1. and Langmuir model (1:1). K_D_ = K_d_/K_a_.

**Table 2 ijms-26-10560-t002:** Binding of ΔFN3.1 *B. longum* GT15 to IL-6, IL-8, and IL-10 at neutral and alkaline pH.

ΔFN3.1	K_a_ (1/Ms)	K_a_ (1/Ms)	K_a_ (1/Ms)
**pH 7.4**			
IL-6	Below the level of detection		
IL-8	330 × 10^−5^	1.6 × 10^−5^	4.9 ± 0.2
IL-10	22.5 × 10^−5^	140 × 10^−5^	62.2 ± 3.1
**pH 8.0**			
IL-6	Below the level of detection		
IL-8	397 × 10^−5^	1.6 × 10^−5^	4.0 ± 0.2
IL-10	Below the level of detection		

**Table 3 ijms-26-10560-t003:** Binding of ΔFN3.3 *B. bifidum* 791 to IL-6, IL-8, and IL-10 at neutral and alkaline pH.

ΔFN3.1	K_a_ (1/Ms)	K_a_ (1/Ms)	K_a_ (1/Ms)
**pH 7.4**			
IL-6	Below the level of detection		
IL-8	878 × 10^−5^	2.0 × 10^−5^	2.3 ± 0.1
IL-10	Below the level of detection		
**pH 8.0**			
IL-6	Below the level of detection		
IL-8	879 × 10^−5^	1.1 × 10^−5^	1.2 ± 0.04
IL-10	Below the level of detection		

**Table 4 ijms-26-10560-t004:** Putative amino acid residues in FN3 domains.

Protein	Amino Acid Residues	Location
ΔFN3.1	Trp78, Ser79, Pro81, Ser82	Cytokine receptor motif (FN3- domain I)
	Trp174, Ser175, Glu177, Ser178	Cytokine receptor motif (FN3- domain II)
	Ala43, Ala51, Thr111, Pro417, Ala424	[31]
ΔFN3.3	Trp77, Ser78, Pro80, Ser81	Cytokine receptor motif (FN3- domain I)
	Glu172, Gly173, Pro175, Ser176	Cytokine receptor motif (FN3- domain II)

## Data Availability

The original contributions presented in this study are included in the article/Appendix A. Further inquiries can be directed to the corresponding authors.

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
