# Peer review of "Differential Binding of ΔFN3 Proteins of Bifidobacterium longum GT15 and Bifidobacterium bifidum 791 to Cytokines Determined by Surface Plasmon Resonance and De Novo Molecular Modeling"

_ijms, 2025, doi:10.3390/ijms262110560_

Round 1

Reviewer 1 Report

Comments and Suggestions for Authors

Authors detected the different bind ability of ΔFN3 proteins of Bifidobacterium longum GT15 and Bifidobacterium bifidum 791 to cytokines determined by surface plasmon resonance and de Novo molecular modeling. However, the novelty and significant is not enough. The authors didn’t present the significance due to the different bind ability. The description of the results is not clear. The describe of FN3 proteins is confused. In most of time, the symbol of Δ represent deleted genes. FN3 used for both the protein and the region. ΔFN3.2 not was used for compared assay. Most of data base on prediction, and most of the  molecular modeling results actually should be move to methods. The graph and table are very primitive.

Author Response

Comments 1: “Novelty and significance is not enough. The authors didn’t present the significance due to the different bind ability.”

Response 1: We appreciate this important remark. In the revised manuscript, we have significantly expanded the discussion of the biological relevance of the observed differential binding. Specifically:

A new paragraph has been added to the Introduction (p. 6, lines 100-110) explaining how strain-specific interactions between microbiota-derived proteins and host cytokines may underlie personalized immune responses.

The Discussion (pp. 1-3, lines 399-425; pp.10-12, lines 488-507 and pp.13, lines 508-516) now includes a dedicated section highlighting the potential of ΔFN3 proteins as strain-specific biomarkers for probiotic selection and as scaffolds for engineered immunomodulators.”

Comments 2: “The description of the results is not clear. The describe of FN3 proteins is confused. In most of time, the symbol of Δ represent deleted genes. FN3 used for both the protein and the region. ΔFN3.2 not was used for compared assay.”

Response 2: We agree that the notation required clarification. The following changes have been made:

In the Introduction (p. 5, lines 85-87), we now explicitly state: “The symbol ‘Δ’ denotes a truncated recombinant fragment (not a gene deletion) encompassing two FN3 domains and the C-terminal region (aa 1494–1994).”

In Section 2.2 (p. 2, lines 162-163), we added: “The criterion for selecting proteins for further research is the possibility of obtaining target proteins in preparative quantities in a native and homogeneous state. (see Section 2.2 and Figure S2). All comparative analyses were performed using ΔFN3.1 (B. longum) and ΔFN3.3 (B. bifidum).”

Comments 3:Most of data base on prediction... molecular modeling results actually should be move to methods. The graph and table are very primitive.”

Response 3: Thank you for this valuable suggestion. We have:

Moved the entire description of structure prediction methodology (including AlphaFold2, RoseTTAFold, I-TASSER, and IntFOLD5 scoring) from Section 2.4 to Section 4.5.1 (Materials and Methods, lines 588-603).

Replaced Figures 1–5 with high-resolution versions (600 dpi), added clear domain labels (FN3-I, FN3-II, C-term), color-coded epitopes, and uniform font (Arial 10 pt).

4. Additional clarifications

We are grateful for critical comments. The novelty of our study is the set of experimental and computational data on how the microbiota interacts with the host organism to establish the functionally competent immune system. The significance of differential binding of the bifidobacterial to individual cytokines is exactly the decisive argument in support of the specificity of microbiota-host interaction rather than a trivial protein-protein complexation. We now revealed the detailed conditions of ΔFN3-cytokine binding. Their role can also differ for individual partners, providing a step forward in understanding the mechanisms of cytokine selection.

This selectivity (or preference) was further substantiated by molecular modeling which generated totally new structures because this question has not been investigated prior to our study. De novo modeling was a challenge since a limited data on spatial structures of bifidobacterial proteins was available. Next, we used a cell-based assay to demonstrate, for the first time, that FN3-cytokine interactions can, again, differ in their physiological outcome. These complexes appeared not to promiscuously influence the expression of cytokine genes but are target-(or gene)-specific. In the case of FN3-TNFα complexes, this interaction can be harmful since the bifidobacterial protein synergized with the cytokine in TNFα gene activation.

The detailed description of FN3 domains has been presented by other authors and us in a couple of citations (see Introduction). Herein we went on to investigate these proteins as structural and functional ligands for cytokines.

The symbol can be applied to the genes and proteins as well. We revised the text to keep the notation consistent. The ΔFN3.2 protein was not tested because we failed to obtain it in a soluble form. Nevertheless, the procedures of gene cloning and protein isolation were, in our opinion, important for the experts who investigate the Bifidobacteria genus. By the way, here we again encounter differential features of FN3 family proteins, i.e., their individual physico-chemical properties.

We admit that the in silico simulations provide evidence that needs to validated by experimental approaches, most importantly by X-ray analyses of crystal structures of FN3-cytokine complexes. This is our next direction of the work. However, at the present stage these simulations were necessary as they generated the initial valuable information about poorly investigated bifidobacterial proteins.

Reviewer 2 Report

Comments and Suggestions for Authors

The manuscript, 'Differential Binding of ΔFN3 Proteins of Bifidobacterium longum GT15 and Bifidobacterium bifidum 791 to Cytokines Determined by Surface Plasmon Resonance and de Novo Molecular Modeling,' by Alekseeva et al, presents experimental and computational analysis of the binding between two proteins of medicinal importance. The work overall seems quite good and is clearly and logically presented. I have only a few points to raise before the manuscript would be ready for publication. (I am only reviewing the computational portion as the other parts are not in my field of expertise).

  1. The discuss of the structures are the beginning of section 2.4 is confusing. The authors state that they obtained the protein structures by homology modelling, and that they were scored by alphafold, rosettafold, etc. Is this in fact the case? or were the structures calculated with alphafold, rosettafold, etc? This seems to be the case, but that is not how it is presented by the authors.
  2. The introduction does not discuss the computational methods at all. I think some discussion of the pros and cons of alphafold versus rosettafold, etc would improve the context for this work immensely. 
  3. The plots from the MD simulations could be seen as mostly not reaching equilibrium, and the text only refers to it qualitatively, Could more quantitative discussion of the equilibration of the MD calculations be included?

Author Response

Comment 1: “The authors state that they obtained the protein structures by homology modelling... Is this in fact the case? Or were the structures calculated with alphafold, rosettafold, etc?”

Response 1: We apologize for the misleading. The structures were generated not by homology modeling but by de novo deep learning-based prediction. The text in Section 2.4 (p. 1, lines 209-214) has been revised:

“Prior to our study, the experimentally confirmed 3D structures of bifidobacterial ∆FN3.1 and ∆FN3.3 proteins were unavailable. Therefore, we performed de novo structural prediction using state-of-the-art deep learning servers AlphaFold2, RoseTTAFold, I-TASSER, and IntFOLD5. Importantly, no 212 homology modeling was performed, as suitable structural templates with high sequence 213 identity were absent”.

Comment 2: “The introduction does not discuss the computational methods at all... some discussion of the pros and cons of alphafold versus rosettafold... would improve the context.”

Response 2:  We added a new paragraph to Introduction (p. 6, lines 100-109) containing the comparison of methodologies:

“The former approach demon-100 strates an exceptional accuracy in predicting global folds of single and multi-domain 101 proteins, especially when evolutionary information is abundant. RoseTTAFold, provides 102 superior modeling of inter-domain arrangements and flexible linkers due to its 103 three-track neural network architecture that simultaneously processes sequence, dis-104 tance, and coordinate information. In contrast, I-TASSER [35–37] relies on threading and 105 fragment assembly, which may limit its performance for proteins lacking close homologs. 106 Given the absence of experimentally resolved structures for bifidobacterial FN3 proteins, 107 we employed a multi-platform strategy to ensure the robustness of our models and to 108 cross-validate the predictions.”

Comment 3: “The plots from the MD simulations could be seen as mostly not reaching equilibrium... Could more quantitative discussion of the equilibration... be included?”

Response 3: Thank you for pointing this out. We now provide quantitative criteria for MD equilibration in Section 2.4 (p. 4, lines 237-244):

“To ensure 237 that MD trajectories reached equilibrium, we applied the quantitative criteria: (i) the 238 backbone RMSD plateaued (fluctuations ≤ 0.2 Å over the last 20 ns), (ii) the radius of 239 gyration stabilized, and (iii) the Ramachandran outliers remained below 1%. For 240 ΔFN3.3-RF, RMSD continued to drift beyond 100 ns (Figure S10C), indicating a failure to 241 equilibrate; this model was excluded from further analysis. In contrast, ΔFN3.1-AF2 and 242 ΔFN3.3-IF5 achieved stable RMSD plateaus (1.8 ± 0.3 Å and 10.5 ± 1.2 Å, respectively), 243 confirming convergence.”

Thank you very much for considering our revised manuscript. We look forward to your feedback.

Round 2

Reviewer 1 Report

Comments and Suggestions for Authors

The author did not make substantial improvements or responses. More wet experimental results are needed to validate these predicted results.

Author Response

Comments 1: “Novelty and significance is not enough. The authors didn’t present the significance due to the different bind ability.”

Response 1: We appreciate this important remark. In the revised manuscript, we have significantly expanded the discussion of the biological relevance of the observed differential binding. Specifically:

A new paragraph has been added to the Introduction (page 3, par. 6, lines 99-108) explaining how strain-specific interactions between microbiota-derived proteins and host cytokines may underlie personalized immune responses.

The Discussion (page 13, par. 1-3, lines 395-420; pages 14-15, par.10-12, lines 483-502 and page 15, par.13, lines 503-511) now includes a dedicated section highlighting the potential of ΔFN3 proteins as strain-specific biomarkers for probiotic selection and as scaffolds for engineered immunomodulators.”

Comments 2: “The description of the results is not clear. The describe of FN3 proteins is confused. In most of time, the symbol of Δ represent deleted genes. FN3 used for both the protein and the region. ΔFN3.2 not was used for compared assay.”

Response 2: We agree that the notation required clarification. The following changes have been made:

In the Introduction (page 2, par. 5, lines 84-86), we now explicitly state: “The symbol ‘Δ’ denotes a truncated recombinant fragment (not a gene deletion) encompassing two FN3 domains and the C-terminal region (aa 1494–1994).”

In Section 2.2 (page 4, par. 2, lines 160-161), we added: “The criterion for selecting proteins for further research is the possibility of obtaining target proteins in preparative quantities in a native and homogeneous state. (see Section 2.2 and Figure S2). All comparative analyses were performed using ΔFN3.1 (B. longum) and ΔFN3.3 (B. bifidum).”

Comments 3:Most of data base on prediction... molecular modeling results actually should be move to methods. The graph and table are very primitive.”

Response 3: Thank you for this valuable suggestion. We have:

Moved the entire description of structure prediction methodology (including AlphaFold2, RoseTTAFold, I-TASSER, and IntFOLD5 scoring) from Section 2.4 to Section 4.5.1 (Materials and Methods, page 17, par. 2-3, lines 582-597).

Replaced Figures 1–5 with high-resolution versions (600 dpi), added clear domain labels (FN3-I, FN3-II, C-term), color-coded epitopes, and uniform font (Arial 10 pt).

4. Additional clarifications

We are grateful for critical comments. The novelty of our study is the set of experimental and computational data on how the microbiota interacts with the host organism to establish the functionally competent immune system. The significance of differential binding of the bifidobacterial to individual cytokines is exactly the decisive argument in support of the specificity of microbiota-host interaction rather than a trivial protein-protein complexation. We now revealed the detailed conditions of ΔFN3-cytokine binding. Their role can also differ for individual partners, providing a step forward in understanding the mechanisms of cytokine selection.

This selectivity (or preference) was further substantiated by molecular modeling which generated totally new structures because this question has not been investigated prior to our study. De novo modeling was a challenge since a limited data on spatial structures of bifidobacterial proteins was available. Next, we used a cell-based assay to demonstrate, for the first time, that FN3-cytokine interactions can, again, differ in their physiological outcome. These complexes appeared not to promiscuously influence the expression of cytokine genes but are target-(or gene)-specific. In the case of FN3-TNFα complexes, this interaction can be harmful since the bifidobacterial protein synergized with the cytokine in TNFα gene activation.

The detailed description of FN3 domains has been presented by other authors and us in a couple of citations (see Introduction). Herein we went on to investigate these proteins as structural and functional ligands for cytokines.

The symbol can be applied to the genes and proteins as well. We revised the text to keep the notation consistent. The ΔFN3.2 protein was not tested because we failed to obtain it in a soluble form. Nevertheless, the procedures of gene cloning and protein isolation were, in our opinion, important for the experts who investigate the Bifidobacteria genus. By the way, here we again encounter differential features of FN3 family proteins, i.e., their individual physico-chemical properties.

We admit that the in silico simulations provide evidence that needs to validated by experimental approaches, most importantly by X-ray analyses of crystal structures of FN3-cytokine complexes. This is our next direction of the work. However, at the present stage these simulations were necessary as they generated the initial valuable information about poorly investigated bifidobacterial proteins.

Reviewer 2 Report

Comments and Suggestions for Authors

Authors have addressed all of my concerns
adequately. I can now recommend the manuscript

for publication.

Author Response

We sincerely thank you for high evaluation of our work, as well as for the constructive comments that allowed us to improve our manuscript.